# CAGroup3D: Class-Aware Grouping for 3D Object Detection on Point Clouds

**Haiyang Wang**[1,6,7]*, **Lihe Ding**[2]*, **Shaocong Dong**[2], **Shaoshuai Shi**[3]†,
**Aoxue Li**[4], **Jianan Li**[2], **Zhenguo Li**[4], **Liwei Wang**[1,5]†

[1]Center for Data Science, Peking University  [2]Beijing Institute of Technology
[3]Max Planck Institute for Informatics  [4]Huawei Noah's Ark Lab, China
[5]National Key Laboratory of General Artificial Intelligence, School of Intelligence Science and
Technology, Peking University  [6]Peng Cheng Laboratory  [7]Pazhou Laboratory (Huangpu)
{wanghaiyang@stu, wanglw@cls}.pku.edu.cn, {dean.dinglihe, shaocong}@bit.edu.cn
sshi@mpi-inf.mpg.de, lijianan15@gmail.com {liaoxue2, Li.Zhenguo}@huawei.com

## Abstract

We present a novel two-stage fully sparse convolutional 3D object detection framework, named CAGroup3D. Our proposed method first generates some high-quality 3D proposals by leveraging the class-aware local group strategy on the object surface voxels with the same semantic predictions, which considers semantic consistency and diverse locality abandoned in previous bottom-up approaches. Then, to recover the features of missed voxels due to incorrect voxel-wise segmentation, we build a fully sparse convolutional RoI pooling module to directly aggregate fine-grained spatial information from backbone for further proposal refinement. It is memory-and-computation efficient and can better encode the geometry-specific features of each 3D proposal. Our model achieves state-of-the-art 3D detection performance with remarkable gains of *+3.6%* on ScanNet V2 and *+2.6%* on SUN RGB-D in term of mAP@0.25. Code will be available at https://github.com/Haiyang-W/CAGroup3D.

## 1 Introduction

As a crucial step towards understanding 3D visual world, 3D object detection aims to estimate the oriented 3D bounding boxes and semantic labels of objects in real 3D scenes. It has been studied for a long time in both academia and industry since it benefits various downstream applications, such as autonomous driving [2, 36], robotics [55, 37] and augmented reality [1, 3]. In this paper, we focus on detecting 3D objects from unordered, sparse and irregular point clouds. Those natural characteristics make it more challenging to directly extend well-studied 2D techniques to 3D detection.

Unlike 3D object detection from autonomous driving scenarios that only considers bird's eye view (BEV) boxes [30, 50, 19, 11, 32, 31], most of existing 3D indoor object detectors [27, 38, 22, 5, 52] typically handle this task through a bottom-up scheme, which extracts the point-wise features from input point clouds, and then groups the points into their respective instances to generate a set of proposals. However, the above grouping algorithms are usually carried out in a class-agnostic manner, which abandons semantic consistency within the same group and also ignores diverse locality among different categories. For example, VoteNet [27] learns the point-wise center offsets and aggregates the points that vote to similar semantic-irrelevant local region. Though impressive, as shown in Figure 1, these methods may fail in cluttered indoor scenes where various objects are close but belong to different categories. Also, the object sizes are diverse for different categories, so that a class-agnostic

---

*Equal contribution.
†Corresponding author.

36th Conference on Neural Information Processing Systems (NeurIPS 2022).

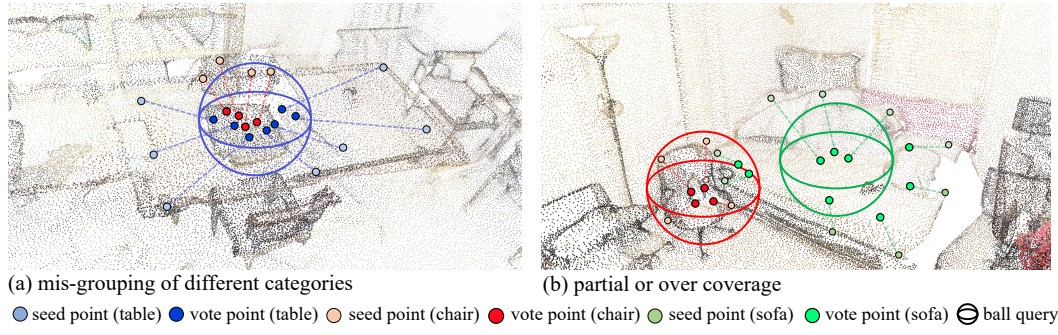

(a) mis-grouping of different categories      (b) partial or over coverage

⬤ seed point (table) ⬤ vote point (table) ⬤ seed point (chair) ⬤ vote point (chair) ⬤ seed point (sofa) ⬤ vote point (sofa) ⊖ ball query

Figure 1: Class-agnostic grouping methods suffer from (a) mis-grouping of different categories within the same local regions, (b) partial coverage of the object surfaces; outliers from the cluttered scene.

local grouping may partially cover the boundary points of large objects and involve more noise outliers for small objects.

Hence, we propose CAGroup3D, a two-stage fully convolutional 3D object detection framework. Our method consists of two novel components. One is the class-aware 3D proposal generation module, which aims to generate reliable proposals by utilizing class-specific local group strategy on the object surface voxels with same semantic predictions. The other one is an efficient fully sparse convolutional RoI pooling module for recovering the features of the missed surface voxels due to semantic segmentation errors, so as to improve the quality of predicted boxes.

Specifically, a backbone network with 3D sparse convolution is firstly utilized to extract descriptive voxel-wise features from raw point clouds. Based on the learned features, we conduct a class-aware local grouping module to cluster surface voxels into their corresponding instance centroids. Different from [27], in order to consider the semantic consistency, we not only shift voxels of the same instance towards the same centroid but also predict per-voxel semantic scores. Given the contiguously distributed vote points with their semantic predictions, we initially voxelize them according to the predicted semantic categories and vote coordinates, so as to generate class-specific 3D voxels for different categories. The voxel size of each category is adaptive to its average spatial dimension. To maintain the structure of fully convolution, we apply sparse convolution as grouping operation centered on each voted voxel to aggregate adjacent voxel features in the same semantic space. Note that these grouping layers are class-dependent but share the same kernel size, thus the larger classes are preferred to be aggregated with larger local regions.

Secondly, given the proposal candidates, fine-grained specific features within 3D proposals need to be revisited from 3D backbone through certain pooling operation for the following box refinement. However, state-of-the-art pooling strategies [31, 8] are memory-and-computation intensive due to the hand-crafted set abstraction [25]. Besides that, its max-pooling operation also harms the geometry distribution. To tackle this problem, we propose RoI-Conv pooling module, which directly adopts the well-optimized 3D sparse convolutions to aggregate voxel features from backbone. It can encode effective geometric representations with a memory-efficient design for further proposal refinement.

In summary, our contributions are three-fold: 1) We propose a novel class-aware 3D proposal generation strategy, which considers both the voxel-wise semantic consistency within the same local group and the object-level shape diversity among different categories. 2) We present RoI-Conv pooling module, an efficient fully convolutional 3D pooling operation for revisiting voxel features directly from backbone to refine 3D proposals. 3) Our approach outperforms state-of-the-art methods with remarkable gains on two challenging indoor datasets, *i.e.*, ScanNet V2 [6] and SUN RGB-D [33], demonstrating its effectiveness and generality.

## 2   Related Work

**3D Object Detection on Point Clouds.** Detecting 3D objects from point clouds is challenging due to orderless, sparse and irregular characteristics. Previous approaches can be coarsely classified into two

lines in terms of point representations, *i.e.*, the voxel-based methods [54, 46, 32, 45, 31, 50, 32, 9, 40] and the point-based methods [27, 38, 5, 22, 52, 47]. Voxel-based methods are mainly applied in outdoor autonomous driving scenarios where objects are distributed on the large-scale 2D ground plane. They process the sparse point clouds by efficient 3D sparse convolution, then project these 3D volumes to 2D grids for detecting bird's eye view (BEV) bboxes by 2D ConvNet. Powered by PointNet series [25, 29], point-based methods are also widely used to predict 3D bounding bboxes. Most of existing methods are in a bottom-up manner, which extracts the point-wise features and groups them to obtain object features. This pipeline has been a great success for estimating 3D bboxes directly from cluttered and dense 3D scenes. However, due to the hand-crafted point sampling and computation intensive grouping scheme applied in PointNet++ [29], they are difficult to be extended to large-scale point clouds. Hence, we propose an efficient fully convolutional bottom-up framework to efficiently detect 3D bboxes directly from dense 3D point clouds.

**Feature Grouping.** Feature grouping is a crucial step for bottom-up 3D object detectors [27, 38, 22, 5, 52, 35], which clusters a group of point-wise features to generate high-quality 3D bounding boxes. Among the numerous successors, voting-based framework [27] is widely used, which groups the points that vote to the same local region. Though impressive, it doesn't consider the semantic consistency, so that may fail in cluttered indoor scenes where the objects of different classes are distributed closely. Moreover, voting-based methods usually adopt a class-agonistic local region for all objects, which may incorrectly group the boundary points of large objects and involve more noise points for small objects. To address the above limitations, we present a class-aware local grouping strategy to aggregate the points of the same category with class-specific center regions.

**Two-stage 3D Object Detection.** Many state-of-the-art methods considered applying RCNN style 2D detectors to the 3D scenes, which apply 3D RoI-pooling scheme or its variants [30, 32, 8, 31, 48, 44] to aggregate the specific features within 3D proposals for the box refinement in a second stage. These pooling algorithms are usually equipped with set abstraction [25] to encode local spatial features, which consists of a hand-crafted query operation (*e.g.*, ball query [25] or vector query [8]) to capture the local points and a max-pooling operation to group the assigned features. Therefore these RoI pooling modules are mostly computation expensive. Moreover, the max-pooling operation also harms the spatial distribution information. To tackle these problems, we propose RoI-Conv pooling, a memory-and-computation efficient fully convolutional RoI pooling operation to aggregate the specific features for the following refinement.

# 3 Methodology

In this paper, we propose CAGroup3D, a two-stage fully convolutional 3D object detection framework for estimating accurate 3D bounding boxes from point clouds. The overall architecture of CAGroup3D is depicted in Figure 2. Our framework consists of three major components: an efficient 3D voxel CNN with sparse convolution as the backbone network for point cloud feature learning (§3.1), a class-aware 3D proposal generation module for predicting high quality 3D proposals by aggregating voxel features of the same category within the class-specific local regions (§3.2) and RoI-Conv pooling module for directly extracting complete and fine-grained voxel features from the backbone to revisit the miss-segmented surface voxels and refine 3D proposals. Finally, we formulate the learning objective of our framework in §3.4.

## 3.1 3D Voxel CNN for Point Cloud Feature Learning

For generating accurate 3D proposals, we first need to learn discriminative geometric representation for describing input point clouds. Voxel CNN with 3D sparse convolution [32, 45, 54, 13, 12] is widely used by state-of-the-art 3D detectors thanks to its high efficiency and scalability of converting the point clouds to regular 3D volumes. In this paper, we adopt sparse convolution based backbone for feature encoding and 3D proposal generation.

3D backbone network equipped with high-resolution feature maps and large receptive fields is critical for accurate 3D bounding box estimation and voxel-wise semantic segmentation. The latter is closely related to the accuracy of succeeding grouping module. To maintain these two characteristics, inspired by the success of HRNet series [39, 16, 34] in segmentation community, we implement a 3D voxel bilateral network with dual resolution based on ResNet [15]. For brevity, we refer it as BiResNet. As shown in Figure 2, our backbone network contains two branches. One is the sparse modification of

a. Class-Aware 3D Proposal Generation

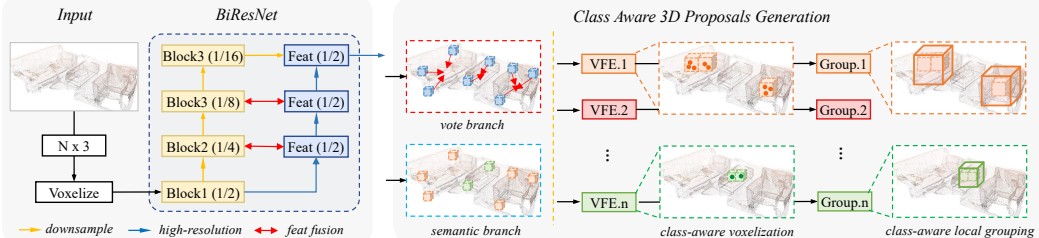

b. RoI-Conv Point Cloud Feature Pooling and Box Refinement

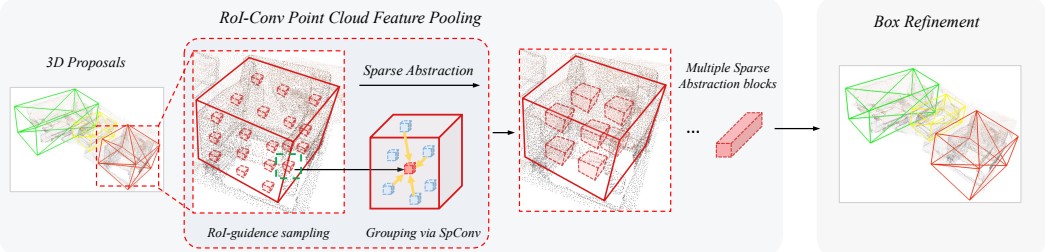

Figure 2: The overall architecture of CAGroup3D. (a) Generate 3D proposals by utilizing class-aware local grouping on the vote space with same semantic predictions. (b) Aggregating the specific features within the 3D proposals by the efficient RoI-Conv pooling module for the following box refinement.

ResNet18 [15] where all 2D convolutions are replaced with 3D sparse convolutions. It can extract multi-scale contextual information with proper downsampling modules. The other one is a auxiliary branch that maintains a high-resolution feature map whose resolution is 1/2 of the input 3D voxels. Specifically, the auxiliary branch is inserted following the first stage of ResNet backbone and doesn't contain any downsampling operation. Similar to [39], we adopt the bridge operation between the two paths to perform the bilateral feature fusion. Finally, the fine-grained voxel-wise geometric features with rich contextual information are generated by the high-resolution branch and facilitate the following module. Experiments also demonstrate that our voxel backbone performs better than previous FPN-based ResNet [20]. More architecture details are in Appendix.

## 3.2 Class-Aware 3D Proposal Generation

Given the voxel-wise geometric features generated by the backbone network, a bottom-up grouping algorithm is generally adopted to aggregate object surface voxels into their respective ground truth instances and generate reliable 3D proposals. Voting-based grouping method [27] has shown great success for 3D object detection, which is performed in a class-agnostic manner. It reformulates Hough voting to learn point-wise center offsets, and then generates object candidates by clustering the points that vote to similar center regions. However, this method may incorrectly group the outliers in the cluttered indoor scenarios (*e.g.*, votes are close together but belong to different categories), which degrades the performance of 3D object detection. Moreover, due to the diverse object sizes of different categories, class-agnostic local regions may mis-group the boundary points of large objects and involve more noise points for small objects.

To address this limitation, we propose the class-aware 3D proposal generation module, which first produces voxel-wise predictions (*e.g.*, semantic maps and geometric shifts), and then clusters the object surface voxels of the same semantic predictions with class-specific local groups.

**Voxel-wise Semantic and Vote Prediction.** After obtaining the voxel features from backbone network, two branches are constructed to output the voxel-wise semantic scores and center offset vectors. Specifically, the backbone network generates a number of $N$ non-empty voxels $\{o_i\}_{i=1}^N$ from backbone, where $o_i = [x_i; f_i]$ with $x_i \in \mathbb{R}^3$ and $f_i \in \mathbb{R}^C$. A voting branch encodes the voxel feature $f_i$ to learn the spatial center offset $\Delta x_i \in \mathbb{R}^3$ and feature offset $\Delta f_i \in \mathbb{R}^C$. Based on the learned spatial and feature offset, we shift voxel $o_i$ to the center of its respective instance and generate vote

point $p_i$ as follow:

$$\{p_i \mid p_i = [x_i + \Delta x_i, f_i + \Delta f_i]\}_{i=1}^N. \tag{1}$$

The predicted offset $\Delta x_i$ is explicitly supervised by a smooth-$\ell_1$ loss with the ground-truth displacement from the coordinate of seed voxel $x_i$ to its corresponding bounding box center.

In parallel with the voting branch, we also construct a semantic branch to output semantic scores $S = \{s_i\}_{i=1}^N$ for all the voxels over $N_{class}$ classes as

$$s_i = \text{MLP}^{\text{sem}}(o_i) \in [0,1]^{N_{class}}, \quad \text{for } i = 1, \cdots, N, \tag{2}$$

where $\text{MLP}^{\text{sem}}(\cdot)$ is a one-layer multi-layer-perceptron (MLP) network and $s_i$ indicates the semantic probability for all classes of voxel $o_i$. We adopt focal loss [21] for calculating voxel segmentation loss to handle the class imbalance issue.

Notably, the vote and semantic targets of each voxel are associated with the ground-truth 3D boxes not the instance or semantic masks, so that it can be easily generalized to the 3D object detection datasets with bounding box annotations. To be specific, for each voxel, only the ground-truth bounding boxes that includes this voxel are selected. Considering the ambiguous cases that a voxel is in multiple ground truth bounding boxes, only the box with the least volume is assigned to this voxel.

**Class-Aware Local Grouping.** This step aims to produce reliable 3D proposals in a bottom-up scheme based on the above voxel-wise semantic scores $\{s_i\}_{i=1}^N$ and vote predictions $\{p_i\}_{i=1}^N$. To carry out grouping with semantic predictions, we first define a score threshold $\tau$ for all categories to individually determine whether a voxel belongs to a category instead of utilizing the one-hot semantic predictions. It can allow the voxel to be associated with multiple classes and thus improve the recall of semantic voxels for each category. Given the semantic scores $S = \{s_i\}_{i=1}^N$, we iterate over all the classes, and slice a point subset from the whole scene of each class that has the score higher than the threshold $\tau$, to form the class-dependent vote set $\{c_j\}_{j=1}^{N_{class}}$,

$$\left\{ c_j \mid c_j = \{p_i : s_i^j > \tau, i = 1, ..., N\} \right\}_{j=1}^{N_{class}}. \tag{3}$$

The above semantic subset is generated in a contiguous euclidean space and the vote points are distributed irregularly. To maintain the structure of pure convolutions and facilitate the succeeding class-aware convolution-based local grouping module, we individually voxelize the vote points in each semantic subset to $\{V_j\}_{j=1}^{N_{class}}$ by employing a voxel feature encoding (VFE) layer with a class-specific 3D voxel size, which is proportional to the class average spatial dimension. Specifically, this re-voxelization process for each class can be formulated as follows:

$$\{v_i\}_{i=1}^{|V_j|} = \text{VFE}(c_j, \ \alpha \cdot d_j, \ Avg), \quad \text{for } j = 1, ..., N_{class}, \tag{4}$$

where $c_j$ is the class-dependent vote set of class index $j$ and $\mid V_j \mid$ is the number of non-empty voxels after class individual re-voxelization. $\alpha \cdot d_j$ is the class-specific voxel size, where $\alpha$ is a predefined scale factor and $d_j = (w_j, h_j, l_j)$ is the category average spatial dimension. $\text{VFE}(\ \cdot \ , \ \alpha \cdot d_j, \ Avg)$ means that the average pooling operation is adopted to voxelize vote features on $j$-th class subset with the voxel size $\alpha \cdot d_j$. Importantly, the voxel size is adaptive among different categories, which is more diverse than the widely used FPN-based prediction structure.

Given the predicted vote voxels of $j$-th class $\{v_i\}_{i=1}^{|V_j|}$, we apply sparse convolutions with a predefined kernel size $k^{(a)}$ on each voxel, and automatically aggregate local context inside the class to generate class-specific geometric features $A^{(j)}$ as follow:

$$A^{(j)} = \{a_i^{(j)} \mid a_i^{(j)} = \text{SparseConv}_{3D}^{(j)}(v_i, \{v_i\}_{i=1}^{|V_j|}, \ k^{(a)})\}_{i=1}^{|V_j|}, \tag{5}$$

where $\text{SparseConv}_{3D}^{(j)}(\cdot_{\text{center}}, \cdot_{\text{support voxels}}, \cdot_{\text{kernel size}})$ is the standard sparse 3D convolution [13] and specific for different classes. A shared anchor-free head is appended to the aggregated features for predicting classification probabilities, bounding box regression parameters and confidence scores.

## 3.3  RoI-Conv point cloud feature pooling for 3D Proposal Refinement

Due to the semantic segmentation errors in stage-I, class-aware local grouping module will mis-group some object surface voxels. So an efficient pooling module is needed to recover the missed voxel

features and also aggregate more fine-grained features from backbone for proposal refinement. State-of-the-art 3D RoI pooling strategies [31, 8] usually adopt set abstraction operation to encode local patterns, which is computation expensive compared to traditional convolution and hand-crafted with lots of hyper-parameters (*e.g.*, radius, the number of neighbors). Moreover, its max-pooling operation also harms the spatial distribution information.

To tackle these limitations and hold a fully convolution structure, we propose RoI-Conv pooling operation, which builds a hierarchical grouping module with well optimized 3D sparse convolutions to directly aggregate RoI-specific features from backbone for further proposal refinement. Our hierarchical structure is composed by a number of *sparse abstraction* block, which contains two key components: the RoI-guidance sampling for selecting a subset of input voxels within the 3D proposals, which defines the centroids of local regions, and a shared sparse convolution layer for encoding local patterns into feature vectors.

**Sparse Abstraction.** As shown in Figure 2, the inputs to this block are a number of $|\mathcal{I}|$ input voxels $\mathcal{I} = \{l_n\}_{n=1}^{\mathcal{I}}$ and $|\mathcal{M}|$ proposals $\mathcal{M} = \{\rho_m\}_{m=1}^{\mathcal{M}}$, where $\rho_m$ is the proposal parameters for guiding the point sampling. The output is a number of $|\mathcal{Q}|$ pooling RoI-specific voxels $\mathcal{Q} = \{q_k\}_{k=1}^{|\mathcal{Q}|}$.

Specifically, given the input voxels and proposals, instead of directly sampling from the whole input space, we adopt the RoI-guidance sampling to uniformly sample $G_x \times G_y \times G_z$ grid points within each 3D proposal in voxel space, which are denoted as $\mathcal{G} = \{g_k \in \mathbb{Z}^3\}_{k=1}^{G_x \times G_y \times G_z \times \mathcal{M}}$. $G_x \times G_y \times G_z$ is the proposal sampling resolution, which are the hyper-parameters independent on proposal sizes. Considering the overlap of different proposals, we merge the repeated grid points and generate a unique points set $\widetilde{\mathcal{G}} = \{g_k\}_{k=1}^{|\widetilde{\mathcal{G}}|}$, where $|\widetilde{\mathcal{G}}|$ is the number of unique grid points. Then, with the RoI-specific points set $\widetilde{\mathcal{G}}$, we exploit a sparse convolution centered on each sampled point to cover a set of neighboring input voxels (*e.g.*, $\mathcal{N}_k = \{l_k^1, l_k^2, ..., l_k^{L_k}\}$ for $g_k$) within the kernel size $k^{(p)}$ as:

$$\mathcal{Q} = \left\{ q_k \mid q_k = \begin{cases} \text{SparseConv}_{3D}(g_k, \mathcal{N}_k, k^{(p)}), & \text{if } L_k > 0, \\ \phi, & \text{if } L_k = 0, \end{cases} \text{ for } k = 1, ..., |\widetilde{\mathcal{G}}| \right\} \quad (6)$$

where $L_k$ is the number of neighboring voxels queried by the $k$-th RoI-specific point with kernel size $k^{(p)}$. $\text{SparseConv}_{3D}(\cdot_{\text{center}}, \cdot_{\text{support voxels}}, \cdot_{\text{kernel size}})$ is a shared sparse 3D convolution for all the proposals and only applied on the points that their neighboring voxel sets are non-empty. $\phi$ means empty voxel. Then, to hold the surface geometry and reduce computation cost, we abandon the empty voxels and output the RoI-specific voxels set $\mathcal{Q}$.

**RoI-Conv Pooling Module.** Our pooling network is equipped with two-layers sparse abstraction block and progressively abstracts the voxel features within each 3D proposal from the backbone to RoI-specific features iteratively as:

$$\begin{aligned} \mathcal{Q}^1 &= \text{SparseAbs}(\mathcal{I}, \{\rho_m\}_{m=1}^{\mathcal{M}}, 7 \times 7 \times 7, 5), \\ \mathcal{F} &= \text{SparseAbs}(\mathcal{Q}^1, \{\rho_m\}_{m=1}^{\mathcal{M}}, 1 \times 1 \times 1, 7), \end{aligned} \quad (7)$$

where $\text{SparseAbs}(\cdot_{\text{input voxels}}, \cdot_{\text{proposals}}, \cdot_{\text{sampling resolution}}, \cdot_{\text{kernel size}})$ denotes our sparse abstraction block. Notably, to encode the voxel features of oriented proposals, we follow the transformation strategy in [32] before the last block, and normalize the input voxels belonging to each proposal to its individual canonical systems. With the RoI feature $\mathcal{F} \in R^C$ of each proposal, the refinement network predicts the size and location (*i.e.*, bbox dimension, center and orientation) residuals relative to the proposal in Stage-I and the targets are encoded by the traditional residual-based method [31, 32].

### 3.4 Learning Objective

Our proposed approach is trained from scratch with semantic loss $\mathcal{L}_{\text{sem}}$, voting loss $\mathcal{L}_{\text{vote-reg}}$, centerness loss $\mathcal{L}_{\text{cntr}}$, bounding box estimation loss $\mathcal{L}_{\text{box}}$, classification losses $\mathcal{L}_{\text{cls}}$ for Stage-I and bbox refinement loss $\mathcal{L}_{\text{rebox}}$ for Stage-II, which are formulated as follows:

$$\begin{aligned} L = &\beta_{\text{sem}}\mathcal{L}_{\text{sem}} + \beta_{\text{vote}}\mathcal{L}_{\text{vote}} + \beta_{\text{cntr}}\mathcal{L}_{\text{cntr}} \\ &+ \beta_{\text{box}}\mathcal{L}_{\text{box}} + \beta_{\text{cls}}\mathcal{L}_{\text{cls}} + \beta_{\text{rebox}}\mathcal{L}_{\text{rebox}}. \end{aligned} \quad (8)$$

$\mathcal{L}_{\text{sem-cls}}$ is a multi-class focal loss [21] used to supervise voxel-wise semantic segmentation. $\mathcal{L}_{\text{vote}}$ is a smooth-$\ell_1$ loss for predicting the center offset of each voxel. In term of the 3D proposal generation

Table 1: 3D detection results on ScanNet V2 [6] and SUN RGB-D [33]. The main comparison is based on the best results of multiple experiments, and the average value of 25 trials is given in brackets. * means the multi-sensor approaches that use both point clouds and RGB images.

| Methods | Presened at | ScanNet V2 | | SUN RGB-D | |
|---|---|---|---|---|---|
| | | mAP@0.25 | mAP@0.5 | mAP@0.25 | mAP@0.5 |
| F-PointNet [26]* | CVPR'18 | 19.8 | 10.8 | 54.0 | - |
| GSPN [49]* | CVPR'19 | 30.6 | 17.7 | - | - |
| 3D-SIS [17]* | CVPR'19 | 40.2 | 22.5 | - | - |
| ImVoteNet [28]* | CVPR'20 | - | - | 63.4 | - |
| TokenFusion [41]* | CVPR'22 | 70.8(69.8) | 54.2(53.6) | 64.9(64.4) | 48.3(47.7) |
| VoteNet [27] | ICCV'19 | 58.6 | 33.5 | 57.7 | - |
| 3D-MPA [10] | CVPR'20 | 64.2 | 49.2 | - | - |
| HGNet [4] | CVPR'20 | 61.3 | 34.4 | 61.6 | - |
| MLCVNet [42] | CVPR'20 | 64.5 | 41.4 | 59.8 | - |
| GSDN [14] | ECCV'20 | 62.8 | 34.8 | - | - |
| H3DNet [52] | ECCV'20 | 67.2 | 48.1 | 60.1 | 39.0 |
| BRNet [5] | CVPR'21 | 66.1 | 50.9 | 61.1 | 43.7 |
| 3DETR [24] | ICCV'21 | 65.0 | 47.0 | 59.1 | 32.7 |
| VENet [43] | ICCV'21 | 67.7 | - | 62.5 | 39.2 |
| Group-free [22] | ICCV'21 | 69.1(68.6) | 52.8(51.8) | 63.0(62.6) | 45.2(44.4) |
| RBGNet [38] | CVPR'22 | 70.6(69.6) | 55.2(54.7) | 64.1(63.6) | 47.2(46.3) |
| HyperDet3D [53] | CVPR'22 | 70.9 | 57.2 | 63.5 | 47.3 |
| FCAF3D [7] | ECCV'22 | 71.5(70.7) | 57.3(56.0) | 64.2(63.8) | 48.9(48.2) |
| Ours | - | **75.1(74.5)** | **61.3(60.3)** | **66.8(66.4)** | **50.2(49.5)** |

module, we follow the same loss functions $\mathcal{L}_{\text{cntr}}$, $\mathcal{L}_{\text{box}}$ and $\mathcal{L}_{\text{cls}}$ defined in [7] to optimize object centerness, bounding box estimation and classification respectively. For the second stage, $\mathcal{L}_{\text{rebox}}$ is the residual-based smooth-$\ell_1$ box regression loss for 3D box proposal refinement, which contains size, box center and angle refinement loss. Besides that, we also add the same IoU loss $\mathcal{L}_{\text{iou}}$ as used in stage-I, and the final box refinement loss is as follows

$$\mathcal{L}_{\text{rebox}} = \sum_{r \in \{x,y,z,l,h,w,\theta\}} \mathcal{L}_{\text{smooth-}\ell_1}(\Delta r^*, \Delta r) + \mathcal{L}_{\text{iou}}, \tag{9}$$

where $\Delta r$ is the predicted residual and $\Delta r^*$ is the corresponding ground truth. The detailed balancing factors are in Appendix.

## 4 Experiments

### 4.1 Datasets and Evaluation Metric

Our CAGroup3D is evaluated on two indoor challenging 3D scene datasets, *i.e.*, ScanNet V2 [6] and SUN RGB-D [33]. For all datasets, we follow the standard data splits adopted in [27].

**ScanNet V2** contains richly-annotated 3D reconstructed indoor scenes with axis-aligned bounding box for most common 18 object categories. It contains 1201 training samples and the remaining 312 scans are left for validation. We follow [27] to sample point clouds from the reconstructed meshes.

**SUN-RGB-D** is a single-view indoor dataset which consists of 10,355 RGB-D images for 3D scene understanding. It contains ∼5K training images annotated with the oriented 3D bounding boxes and the semantic labels for 10 categories. To feed the point data to our method, we follow [27] and convert the depth images to point clouds using the provided camera parameters.

All the experiment results on both datasets are evaluated by a standard evaluation protocol [27, 7], which uses mean average precision(mAP) with different IoU thresholds, *i.e.*, 0.25 and 0.50.

Table 2: Effect of Semantic Prediction, Diverse Local Group, RoI-Conv and BiResNet.

| Semantic Prediction | Diverse Local Group | BiResNet | RoI-Conv | mAP@0.25 | mAP@0.5 |
|:---:|:---:|:---:|:---:|:---:|:---:|
| | | | | 68.22 | 53.17 |
| ✓ | | | | 69.24 | 54.05 |
| ✓ | ✓ | | | 72.10 | 57.07 |
| ✓ | ✓ | ✓ | | 73.21 | 57.18 |
| ✓ | ✓ | ✓ | ✓ | **74.50** | **60.31** |

Table 3: Ablation study of the class-aware local group with different scale factor $\alpha$. ($k^{(a)} = 9$)

| scale factor $\alpha$ | mAP@0.25 | mAP@0.5 |
|:---:|:---:|:---:|
| 1.00 | 36.38 | 24.66 |
| 0.60 | 61.49 | 47.41 |
| 0.20 | 74.21 | 58.77 |
| 0.15 | **74.50** | **60.31** |
| 0.10 | 74.01 | 59.03 |
| 0.05 | 72.98 | 58.10 |

Table 4: Ablation study of the class-aware local group with different semantic threshold $\tau$.

| sem thres $\tau$ | mAP@0.25 | mAP@0.5 |
|:---:|:---:|:---:|
| 0.20 | 73.51 | 59.50 |
| 0.10 | 74.28 | 59.97 |
| 0.08 | 74.38 | 60.10 |
| 0.06 | **74.51** | **60.27** |
| 0.04 | 74.38 | 59.98 |
| 0.02 | 73.53 | 58.85 |

## 4.2 Implementation Details

**Network Architecture Details.** For both datasets, we set the voxel size as $0.02m$. As for the backbone, we use the same 3D voxel ResNet18 introduced in [14] as the downsample branch to extract rich contextual information and set the voxel size of high-resolution branch to $0.04m$. In terms of class-aware 3D proposal generation module, we set the scale factor of re-voxelization $\alpha$ to 0.15. The semantic threshold $\tau$ is initially set to 0.15 and then decreases by 0.02 every 10 epochs for ScanNet V2 and 4 epochs for SUN RGB-D until it reaches the minimum value of 0.05. Moreover, the kernel size of convolution-based grouping module $k^{(a)}$ is 9. We stack two sparse abstraction blocks for extracting RoI-specific representations, in where the proposal sampling resolutions are set to $7 \times 7 \times 7$ and $1 \times 1 \times 1$, and the kernel sizes of sparse grouping $k^{(p)}$ are set to 5 and 7 respectively.

**Training and Evaluation Scheme.** Our model is trained in an end-to-end manner by AdamW optimizer [23]. Following the [7], we set batch size, initial learning rate and weight decay are 16, 0.001 and 0.0001 for both datasets. Training ScanNet V2 requires 120 epochs with the learning rate decay by 10x on 80 epochs and 110 epochs. SUN RGB-D takes 48 epochs and learning rate decayed on 32, 44 epochs. All models are trained on two NVIDIA Tesla V100 GPUs with a 32 GB memory per-card. The gradnorm clip [51, 18] is applied to stabilize the training dynamics. We follow the same evaluation scheme in [22], which runs training for 5 times and test each trained model for 5 times. We report both the best and average metrics across all results.

## 4.3 Benchmarking Results

We report the comparison results with state-of-the-art 3D detection methods on ScanNet V2 [6]] and SUN RGB-D [33] benchmark. Same as [22], we report the best and average results of 5×5 trials.

As shown in Table 1, our approach leads to 75.1 in terms of mAP@0.25 and 61.3 in terms of mAP@0.50 on ScanNet V2 [6], which is +3.6 and +4.0 better than the state-of-the-art [7]. For SUN RGB-D, our approaches achieves 66.8 and 50.2, which gains +2.6 and +1.3 in terms of mAP, with 3D IoU threshold 0.25 and 0.5 respectively. Moreover, our approach outperforms all the multi-sensor based methods, which is more challenging for SUN RGB-D due to its relatively poor point clouds.

## 4.4 Ablation Studies and Discussions

We conduct extensive ablation studys on the *val* sets of ScanNet V2 to analyze individual components of our proposed method. Following [22], we report the average performance of 25 trials by default.

Table 5: Comparison with other RoI pooling approaches.

| RoI Method | mAP@0.25 | mAP@0.5 | memory |
|---|---|---|---|
| PointRCNN [30] | 73.65 | 57.83 | 8,054MB |
| Part-A$^2$ [32] | 74.01 | 58.89 | 6,540MB |
| Ours-SA [29] | 73.89 | 58.14 | 11,508MB |
| Ours-SpConv | **74.50** | **60.31** | **2,468MB** |

Table 6: Ablate the depth of RoI module.

| $G_t, t = 1...$ | mAP@0.25 | mAP@0.5 |
|---|---|---|
| $\{1\}$ | 72.33 | 58.42 |
| $\{7, 1\}$ | **74.50** | 60.31 |
| $\{7, 5, 1\}$ | 73.91 | **60.61** |
| $\{7, 5, 3, 1\}$ | 73.72 | 59.50 |

Table 7: Effect of different proposal sampling resolution $G_x \times G_y \times G_z$ in the $1^{st}$ sparse abstraction.

| $G_x \times G_y \times G_z$ | mAP@0.25 | mAP@0.5 |
|---|---|---|
| $3 \times 3 \times 3$ | 73.56 | 59.61 |
| $5 \times 5 \times 5$ | 74.21 | 60.09 |
| $7 \times 7 \times 7$ | **74.50** | 60.31 |
| $9 \times 9 \times 9$ | 74.41 | 60.43 |
| $11 \times 11 \times 11$ | 74.44 | **60.49** |

Table 8: Effect of using different grouping kernel sizes $k^{(p)}$ in the $1^{st}$ sparse abstraction.

| $k^{(p)}$ | mAP@0.25 | mAP@0.5 |
|---|---|---|
| $3 \times 3 \times 3$ | 73.87 | 59.82 |
| $5 \times 5 \times 5$ | 74.50 | 60.31 |
| $7 \times 7 \times 7$ | 74.47 | 60.39 |
| $9 \times 9 \times 9$ | **74.52** | 60.42 |
| $11 \times 11 \times 11$ | 74.45 | **60.46** |

**Effect of class-aware local grouping module.** We first ablate the effects of class-aware local grouping module in Table 2, 3, 4. In Table 2, the base competitor ($1^{st}$ row) is the fully sparse convolutional VoteNet [27] we implemented. Compared with CAGroup3D, it abandons the two stage refinement, replaces the BiResNet with FPN-based variant and groups the vote voxels with the voxel size of 0.02 in a class-agnostic manner. As evidenced in the $1^{st}$ and the $2^{nd}$ rows, by considering the semantic consistency, our model performs better, *i.e.*, $68.22 \rightarrow 69.24$, $53.17 \rightarrow 54.05$ on mAP@0.25 and mAP@0.5. Combining the semantic predictions with class-specific local groups ($3^{rd}$ rows), our model achieves a significant improvement, *i.e.*, $68.22 \rightarrow 72.10$, $53.17 \rightarrow 57.07$. That verifies our motivation that semantic consistency within the same group and diverse locality among different categories are crucial for an effective grouping algorithm.

Our class-aware local grouping module also works well for a range of hyper-parameters, such as the scale factor $\alpha$ of class-specific re-voxelization and the semantic threshold $\tau$. Table 3 shows the performance of our model with different scale factors. We first set $k^{(a)}$ to 9 and reduce the scale factor $\alpha$ gradually. With the decrease of $\alpha$, the output voxels are more fine-grained, which is beneficial to estimate accurate bounding boxes and localize small objects. However, a very small voxel size will degrade the performance. With the fixed $k^{(a)}$, smaller voxel sizes will lead to smaller local regions, which may mis-group the boundary object points, and thus it is hard for the network to accurately capture local object geometry. Table 4 also ablates the effectiveness of different semantic threshold. We observe that the performance gradually improves with the decrease of $\tau$. However, too small threshold can also drops the performance due to the abandonment of semantic consistency.

**Effect of RoI-Conv pooling module.** Table 2 demonstrates the effectiveness of two-stage refinement with RoI-Conv pooling module. By comparing the $4^{rd}$ and $5^{th}$ rows, we find that our refinement module can really help to detect more accurate 3D bounding boxes, especially in term of mAP@0.50, *i.e.*, $57.18 \rightarrow 60.31$. To further ablate the high performance of our RoI-Conv pooling module, we compare it to several pooling strategies [30, 32] widely used in 3D object detection. For a fair comparison, we only switch the RoI pooling algorithm while all other settings remain unchanged, (*e.g.*, class-aware local grouping and BiResNet). Table 5 shows that our approach surpasses others on the performance of both detection scores and computation cost with a remarkable margin. Move details about the above competitors are in Appendix. Table 6 shows the impact of stacking different number of sparse abstraction blocks with $k^{(p)} = 5$ except the last layer. We choose the relative shallow design of two layers as there are no obvious improvement by additional deepening. We further ablate the influence of proposal sampling resolution and sparse kernel size $k^{(p)}$ based on our two layers architecture. Table 7 and 8 show that both the larger $G$ and $k^{(p)}$ can capture more fine-grained geometric details and lead to better performance. Considering the trade-off between memory usage and performance improvement, our model finally sets $G$ and $k^{(p)}$ of the $1^{st}$ layer to $7 \times 7 \times 7$ and 5.

**Effect of bilateral feature learning.** In Table 2, we investigate the effects of bilateral backbone (BiResNet18) by replacing it with FPN-based ResNet18 [7, 14]. The $3^{th}$ and $4^{th}$ rows exhibit that the

performance drops, especially in term of mAP@0.25, from 73.21 → 72.10, when adopts FPN-based backbone. This phenomenon validates that our bilateral backbone could learn much richer contextual information while maintain the high-resolution representations.

## 5  Conclusion

In this paper, we propose CAGroup3D, a two-stage fully convolutional 3D object detector, which generates some 3D proposals by utilizing the class-aware local grouping module on the object voxels with same semantic predictions. Then, to efficiently recover the features of the missed voxels due to incorrect semantic segmentation, we design a fully sparse convolutional RoI pooling module, which is memory-and-computation efficient and could better encode spatial information than previous max-pooling based RoI methods. Equipped with the above designs, our model achieves state-of-the-art on ScanNet V2 and SUN RGB-D benchmarks with remarkable performance gains.

**Limitations.** CAGroup3D mainly focuses on the inter-category locality, which is class-specific and diverse among the different classes, but ignores the intra-category discriminations. Due to the incompleteness of the point cloud and the scale variance within the classes, the object spatial dimension of the same class is also variational, which leads to the diverse intra-category locality. Although our grouping algorithm can implicitly handle this problem in some degree by the learnable convlutional aggregation module, it is still an open problem and will be studied in the future.

## Acknowledgement

Liwei Wang is supported by National Science Foundation of China (NSFC62276005), The Major Key Project of PCL (PCL2021A12), Exploratory Research Project of Zhejiang Lab (No. 2022RC0AN02), and Project 2020BD006 supported by PKUBaidu Fund. We gratefully acknowledge the support of MindSpore[*].

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
