In the supplementary material, we first provide more implementation details of the network architecture (§1), then present the per-category evaluation (§2.1), latency and runtime memory analysis (§2.2), more ablation studies (§2.3) and visualization of quantitative results (§2.4).

# 1 Implementation Details

As mentioned in the main paper, the CAGroup3D architecture consists of a backbone with dual resolution named BiResNet, a class-aware 3D proposal generation module and a RoI-Conv refinement module. We first detail the backbone and proposal generation module, then elaborate the RoI-Conv refinement module as well as its competitors, and finally present the details of our loss functions.

## 1.1 BiResNet Backbone

Our backbone network is built upon MinkowskiEngine [3], an auto-differentiation library for sparse tensors. In all experiments, we voxelize the original point clouds into sparse tensors with a voxel size of 0.02m and feed them into the backbone network. BiResNet contains two branches, one is the sparse modification of ResNet18 [5] to extract pyramid contextual features with proper downsampling modules, the other one is an auxiliary branch to hold a high-resolution feature map whose resolution is 1/2 of the input 3D voxels. To achieve information interaction between the two streams, we construct a bilateral fusion block, which includes fusing the high-resolution branch into the low-resolution (high-to-low) and low-resolution into high-resolution (low-to-high). As for high-to-low fusion, high-resolution features are downsampled by a sparse convolution block with a specifical stride (*e.g.*, 2 and 4 for different stages) before being added to the low-resolution feature map. Meanwhile, an interpolation operation and another channel-compression convolution are used to upsample the low-resolution feature map before being fused with the high-resolution auxiliary branch. All convolution layers are followed by batch or instance normalization and ReLU activation function. The output of the backbone network are 64-dimensional voxel-wise latent features.

## 1.2 Class-aware 3D Proposal Generation Module

The class-aware 3D proposal generation module consists of a semantic and vote prediction module, a class-aware local grouping module and an anchor-free proposal head. The detailed computation procedure is provided in Algorithm 0. The proposal head comprises three parallel sparse convolutional layers with weights shared across all class-individual feature maps. For each candidate object, theses layers output classification probabilities for each class, bounding box parameters and 3D centerness values separately. Finally, we filter out those proposal bounding boxes with score less than 0.01, then apply oriented NMS with 3D IoU threshold of 0.5 to remove overlapped bounding boxes and reduce the number of proposals.

## 1.3 RoI-Conv Refinement Module

Given the proposals of Stage-I, we further select 128 proposals whose 3D IoU with ground truth are greater than 0.3 as training samples for each scene, while reserve all proposals during inference. Finally, the proposals and voxel features from backbone are fed into RoI-Conv pooling module with two stacked sparse abstraction blocks to obtain the RoI-specific features. We also provide more implementation details of other RoI pooling strategies mentioned in the main paper.

**PointRCNN**. We first slightly enlarge the proposals by 0.3m, then randomly take out 128 voxels $\{l_n\}_{n=1}^{128}$ within each proposal for further processing. These cropped voxels are regarded as input points and fed into the hierarchical PointNet++ [9] with two SA layers to obtain the final RoI-specific features. The first SA layer uses farthest point sampling (FPS) to sample 32 key points from the input and applies a set abstraction operation [7] centered on each key points to encode local patterns. The radius and number of neighbors are set to 0.4m and 16. Finally the sampled key points are pooled to a feature vector by the last SA layer for further proposal refinement.

**Part-A$^2$**. Instead of directly processing the irregular points within proposals as PointRCNN, Part-A$^2$ converts the contiguously distributed points into regular voxels with a fixed spatial shape, where the average pooling operation is adopted to pool the points in the same voxel. For a fair comparison, we adopt the same spatial shape (*i.e.*, $7 \times 7 \times 7$) as ours, and then several sparse convolutions are stacked

---

**Algorithm 1** Algorithm of Class-Aware 3D proposal Generation Module.

---

**Input:** Seed voxels $\{o_i\}_{i=1}^N$, semantic threshold $\tau$, kernel sizes of aggregation $k^{(a)}$,
Voxel sizes of different classes $\{d_j\}_{j=1}^{N_{class}}$, scale factor $\alpha$.

**Output:** Proposals $P$

1: Initialize class-aware aggregation results $A = \{\}$.
/*Voxel-wise Semantic and Vote Prediction*/

2: $\{p_i\}_{i=1}^N = \{\text{MLP}^{\text{vote}}(o_i)\}_{i=1}^N$, $\quad \{s_i\}_{i=1}^N = \{\text{MLP}^{\text{sem}}(o_i) \in [0,1]^{N_{class}}\}_{i=1}^N$
/*Class-Aware Local Grouping*/

3: **for** $j \leftarrow 0$ to $N_{class}$ **do**
/*Slice a semantic subset with $\tau$*/

4: $\quad c_j = \{p_i : s_i^j > \tau, i = 1, ..., N\}$
/*Class-Aware Re-voxelization*/

5: $\quad \{v_i\}_{i=1}^{|V_j|} = \text{VFE}(c_j, \ \alpha \cdot d_j, \ Avg)$
/*Class-dependent SpConv Aggregation with $k^{(a)}$*/

6: $\quad A^{(j)} = \{a_i^{(j)} \mid a_i^{(j)} = \text{SparseConv}_{\text{3D}}^{(j)}(v_i, \{v_i\}_{i=1}^{|V_j|}, k^{(a)})\}_{i=1}^{|V_j|}$
/*Merge the subset*/

7: $\quad A$ append $A^{(j)}$ $\qquad\qquad\qquad$ ▷ Not a unique OP that each loc may have multiple features.

8: **end for**
/*Proposal Head with NMS*/

9: $P = \text{NMS}\left(\{\text{MLP}(A_l)\}_{l=0}^{|V_0|+...+|V_{N_{class}}|}\right)$

10: **Return** $P$

---

to aggregate all part features into a feature vector. Notably, we follow the original paper [10] and keep the empty voxels in each proposal to encode the bounding box's geometric information.

**Ours-SA**. In this variant, we replace the sparse convolution operation used for encoding local patterns in our sparse abstraction block with set abstraction [9]. Specifically, given the RoI-specific points set $\widetilde{\mathcal{G}} = \{g_k\}_{k=1}^{|\widetilde{\mathcal{G}}|}$ sampled from the proposals, instead of exploiting sparse convolution centered on each points, we adopt ball query to cover neighboring voxels. Then a PointNet operation is applied on each query group to learn the local patterns. We follow the same two-layers architecture and proposal sampling resolutions as our RoI-Conv module. Their corresponding radius and number of neighbors are set to (0.3m, 2.0m) and $(16, 7 \times 7 \times 7)$ respectively.

As mentioned in the main paper, we compare our RoI-Conv module with the above three variants both on detection scores and computation cost. Note that the computation cost is measured by training memory with the batch size of 8. The experiments show that our RoI-Conv module has significant superiority. All the experiments are run on the same workstation and environment.

We further explain how we change the depth of RoI module introduced in the main paper. To be specific, we stack different number of sparse abstraction blocks with fixed sparse kernel size $k^{(p)} = 5$ and decreasing proposal sampling resolutions $\{G_t\}, t = 1, ..., n$, where $n$ is the number of blocks. For example, $\{G_t\} = \{1\}$ means we only sample one grid point (proposal center) for each proposal and aggregate input voxels from backbone by directly applying sparse convolution centered on these points; $\{G_t\} = \{7, 5, 1\}$ means the first sparse abstraction block outputs a voxel set $\mathcal{Q}^1$ where the voxels are sampled from the proposals with $7 \times 7 \times 7$ resolution and serve as convolution centers to encode their local patterns from the input voxels. The second sparse abstraction block further samples a smaller voxel set $\mathcal{Q}^2$ from the proposals with $5 \times 5 \times 5$ resolution, and similarly obtains the voxelwise output features by applying sparse convolution centered on $\mathcal{Q}^2$ to cover their neighbors from $\mathcal{Q}^1$. Finally, $\mathcal{Q}^2$ is fed into the last sparse abstraction block. We use a sparse convolution with $k^{(p)} = 5$ to aggregate all the part features into the proposal center and get the RoI-specific feature vector. Other settings can be easily understood by analogy with the above explanation. Note that the sparse kernel size $k^{(p)}$ in the last block is equal to the proposal sampling resolution in the second-to-last block to aggregate all information in the proposal.

## 1.4 Loss Function Details

Our model is trained end-to-end with a multi-task loss including semantic loss $\mathcal{L}_{\text{sem}}$, voting loss $\mathcal{L}_{\text{vote-reg}}$, centerness loss $\mathcal{L}_{\text{cntr}}$, bounding box estimation loss $\mathcal{L}_{\text{box}}$, classification losses $\mathcal{L}_{\text{cls}}$ for Stage-I and bbox refinement loss $\mathcal{L}_{\text{rebox}}$ for Stage-II.

$$L = \beta_{\text{sem}}\mathcal{L}_{\text{sem}} + \beta_{\text{vote}}\mathcal{L}_{\text{vote}} + \beta_{\text{cntr}}\mathcal{L}_{\text{cntr}}$$
$$+\beta_{\text{box}}\mathcal{L}_{\text{box}} + \beta_{\text{cls}}\mathcal{L}_{\text{cls}} + \beta_{\text{rebox}}\mathcal{L}_{\text{rebox}}. \tag{1}$$

The second stage loss $\mathcal{L}_{\text{rebox}}$ consists of a regression loss $\mathcal{L}_{\text{smooth-}\ell_1}$ and a iou loss $\mathcal{L}_{\text{iou}}$. For the regression loss, both the 3D proposals and their corresponding ground-truth bounding boxes are transformed into the canonical coordinate systems, which means the 3D proposal $b_i = (x_i, y_i, z_i, h_i, w_i, l_i, \theta_i)$ and ground-truth bounding box $b_i^{gt} = (x_i^{gt}, y_i^{gt}, z_i^{gt}, h_i^{gt}, w_i^{gt}, l_i^{gt}, \theta_i^{gt})$ would be transformed to

$$\tilde{b}_i = (0,\ 0,\ 0,\ h_i,\ w_i,\ l_i,\ 0),$$
$$\tilde{b}_i^{gt} = (x_i^{gt} - x_i,\ y_i^{gt} - y_i,\ z_i^{gt} - z_i,\ h_i^{gt},\ w_i^{gt},\ l_i^{gt},\ \theta_i^{gt} - \theta_i). \tag{2}$$

Then following the traditional residual learning method and sin-cos heading encoding strategy, we obtain the final target $t$ as follow:

$$t = (\frac{x_i^{gt} - x_i}{d},\ \frac{y_i^{gt} - y_i}{d},\ \frac{z_i^{gt} - z_i}{d},\ \log(\frac{h_i^{gt}}{h_i}),\ \log(\frac{w_i^{gt}}{w_i}),\ \log(\frac{l_i^{gt}}{l_i}),\ \sin(\Delta_\theta),\ \cos(\Delta_\theta)), \tag{3}$$

where $d = \sqrt{h_i^2 + w_i^2 + l_i^2}$, $\Delta_\theta = \theta_i^{gt} - \theta_i$. Finally the smooth-$\mathcal{L}_1$ loss is adopted to compute the regression loss. For the iou loss, we get the final refined bounding boxes decoded from the prediction logits and compute their rotated IoU with ground-truth bouding boxes as used in Stage-I.

The balancing factors are set default as $\beta_{sem} = 1.0, \beta_{vote} = 1.0, \beta_{cntr} = 1.0, \beta_{box} = 1.0, \beta_{cls} = 1.0, \beta_{rebox} = 0.5$.

Table 1: 3D detection scores per category on the ScanNetV2, evaluated with mAP@0.25 IoU.

| | cab | bed | chair | sofa | tabl | door | wind | bkshf | pic | cntr | desk | curt | frig | showr | toil | sink | bath | ofurn | mAP |
|---|---|---|---|---|---|---|---|---|---|---|---|---|---|---|---|---|---|---|---|
| VoteNet [8] | 47.87 | 90.79 | 90.07 | 90.78 | 60.22 | 53.83 | 43.71 | 55.56 | 12.38 | 66.85 | 66.02 | 52.37 | 52.05 | 63.94 | 97.40 | 52.32 | 92.57 | 43.37 | 62.90 |
| MLCVNet [11] | 42.50 | 88.50 | 90.00 | 87.40 | 63.50 | 56.90 | 47.00 | 57.00 | 12.00 | 63.90 | 76.10 | 56.70 | 60.90 | 65.90 | 98.30 | 59.20 | 87.20 | 47.90 | 64.50 |
| BRNet [2] | 49.90 | 88.30 | 91.90 | 86.90 | 69.30 | 59.20 | 45.90 | 52.10 | 15.30 | 72.00 | 76.80 | 57.10 | 60.40 | 73.60 | 93.80 | 58.80 | 92.20 | 47.10 | 66.10 |
| H3DNet [12] | 49.40 | 88.60 | 91.80 | 90.20 | 64.90 | 61.00 | 51.90 | 54.90 | 18.60 | 62.00 | 75.90 | 57.30 | 57.20 | 75.30 | 97.90 | 67.40 | 92.50 | 53.60 | 67.20 |
| Group-free [6] | 52.10 | 92.90 | 93.60 | 88.00 | **70.70** | 60.70 | 53.70 | 62.40 | 16.10 | 58.50 | 80.90 | 67.90 | 47.00 | 76.30 | 99.60 | 72.00 | **95.30** | 56.40 | 69.10 |
| FCAF3D [4] | 57.20 | 87.00 | 95.00 | 92.30 | 70.30 | 61.10 | 60.20 | 64.50 | 29.90 | 64.30 | 71.50 | 60.10 | 52.40 | **83.90** | 99.90 | **84.70** | 86.60 | 65.40 | 71.50 |
| Ours | **60.37** | **93.00** | **95.25** | **92.32** | 69.95 | **67.95** | **63.60** | **67.29** | **40.70** | **77.01** | **83.87** | **69.43** | **65.65** | 73.00 | **99.97** | 79.70 | 86.98 | **66.12** | **75.12** |

Table 2: 3D detection scores per category on the ScanNetV2, evaluated with mAP@0.50 IoU.

| | cab | bed | chair | sofa | tabl | door | wind | bkshf | pic | cntr | desk | curt | frig | showr | toil | sink | bath | ofurn | mAP |
|---|---|---|---|---|---|---|---|---|---|---|---|---|---|---|---|---|---|---|---|
| VoteNet [8] | 8.10 | 76.10 | 67.20 | 68.80 | 42.40 | 15.30 | 6.40 | 28.00 | 1.30 | 9.50 | 37.50 | 11.60 | 27.80 | 10.00 | 86.50 | 16.80 | 78.90 | 11.70 | 33.50 |
| BRNet [2] | 28.70 | 80.60 | 81.90 | 80.60 | 60.80 | 35.50 | 22.20 | 48.00 | 7.50 | **43.70** | 54.80 | 39.10 | 51.80 | 35.90 | 88.90 | 38.70 | 84.40 | 33.00 | 50.90 |
| H3DNet [12] | 20.50 | 79.70 | 80.10 | 79.60 | 56.20 | 29.00 | 21.30 | 45.50 | 4.20 | 33.50 | 50.60 | 37.30 | 41.40 | 37.00 | 89.10 | 35.10 | **90.20** | 35.40 | 48.10 |
| Group-free [6] | 26.00 | 81.30 | 82.90 | 70.70 | 62.20 | 41.70 | 26.50 | 55.80 | 7.80 | 34.70 | **67.20** | 43.90 | 44.30 | 44.10 | 92.80 | 37.40 | 89.70 | 40.60 | 52.80 |
| FCAF3D [4] | 35.80 | 81.50 | 89.80 | 85.00 | 62.00 | 44.10 | 30.70 | 58.40 | 17.90 | 31.30 | 53.40 | 44.20 | 46.80 | **64.20** | 91.60 | 52.60 | 84.50 | 57.10 | 57.30 |
| Ours | **41.35** | **82.82** | **90.82** | **85.62** | **64.93** | **54.33** | **37.33** | **64.10** | **31.38** | 41.08 | 63.62 | **44.38** | **56.95** | 49.26 | **98.19** | **55.44** | 82.40 | **58.82** | **61.27** |

Table 3: 3D detection scores per category on the SUN RGB-D, evaluated with mAP@0.25 IoU.

| | bathtub | bed | bookshelf | chair | desk | dresser | nightstand | sofa | table | toilet | mAP |
|---|---|---|---|---|---|---|---|---|---|---|---|
| VoteNet [8] | 75.50 | 85.60 | 31.90 | 77.40 | 24.80 | 27.90 | 58.60 | 67.40 | 51.10 | 90.50 | 59.10 |
| MLCVNet [11] | 79.20 | 85.80 | 31.90 | 75.80 | 26.50 | 31.30 | 61.50 | 66.30 | 50.40 | 89.10 | 59.80 |
| H3DNet [12] | 73.80 | 85.60 | 31.00 | 76.70 | 29.60 | 33.40 | 65.50 | 66.50 | 50.80 | 88.20 | 60.10 |
| BRNet [2] | 76.20 | 86.90 | 29.70 | 77.40 | 29.60 | 35.90 | 65.90 | 66.40 | 51.80 | 91.30 | 61.10 |
| HGNet [1] | 78.00 | 84.50 | **35.70** | 75.20 | 34.30 | 37.60 | 61.70 | 65.70 | 51.60 | 91.10 | 61.60 |
| Group-free [6] | 80.00 | 87.80 | 32.50 | 79.40 | 32.60 | 36.00 | 66.70 | 70.00 | 53.80 | 91.10 | 63.00 |
| FCAF3D [4] | 79.00 | 88.30 | 33.00 | 81.10 | 34.00 | 40.10 | 71.90 | 69.70 | 53.00 | 91.30 | 64.20 |
| Ours | **81.37** | **90.81** | 32.64 | **82.97** | **39.19** | **42.74** | **73.49** | **72.22** | **59.64** | **92.91** | **66.80** |

Table 4: 3D detection scores per category on the SUN RGB-D, evaluated with mAP@0.50 IoU.

| | bathtub | bed | bookshelf | chair | desk | dresser | nightstand | sofa | table | toilet | mAP |
|---|---|---|---|---|---|---|---|---|---|---|---|
| VoteNet [8] | 45.40 | 53.40 | 6.80 | 56.50 | 5.90 | 12.00 | 38.60 | 49.10 | 21.30 | 68.50 | 35.80 |
| H3DNet [12] | 47.60 | 52.90 | 8.60 | 60.10 | 8.40 | 20.60 | 45.60 | 50.40 | 27.10 | 69.10 | 39.00 |
| BRNet [2] | 55.50 | 63.80 | 9.30 | 61.60 | 10.00 | 27.30 | 53.20 | 56.70 | 28.60 | 70.90 | 43.70 |
| Group-free [6] | 64.00 | 67.10 | 12.40 | 62.60 | 14.50 | 21.90 | 49.80 | 58.20 | 29.20 | 72.20 | 45.20 |
| FCAF3D [4] | 66.20 | **69.80** | 11.60 | 68.80 | 14.80 | 30.10 | 59.80 | 58.20 | 35.50 | **74.50** | 48.90 |
| Ours | **68.55** | 67.44 | **13.82** | **70.84** | **17.28** | **30.92** | **59.91** | **61.27** | **39.22** | 72.73 | **50.20** |

## 2 More Results

### 2.1 Per-class Evaluation

We evaluate per-category on ScanNet V2 and SUN RGB-D under different IoU thresholds. Table 1, 2 report the results on 18 classes of ScanNet V2 with 0.25 and 0.5 box IoU thresholds respectively. Table 3, 4 show the results on 10 classes of SUN RGB-D with 0.25 and 0.5 box IoU thresholds. Our approach outperforms the baseline VoteNet [8] and previous state-of-the-art method FCAF3D [4] significantly in almost every category. Notably, our model significantly performs better than prior works on tiny classes (e.g., picture: +10.80 and +13.48 better than the SOTA on ScanNet V2), which demonstrates the effectiveness of our local grouping strategy.

### 2.2 Latency and Memory Analysis.

We also report the latency and memory usage of our CAGroup3D on ScanNet V2. For a fair comparison, we re-measure all the methods on the same workstation (Single NVIDIA RTX 3090 GPU card, 256G RAM, and Xeon(R) E5-2638 v3) and enviroment (Unbuntu-16.04, Python 3.7, Cuda-11.1 and Pytorch-1.8.1). The official code of other methods is used for evaluation. Table 5 shows that our method achieves better performance with a competitive speed. The time cost of CAGroup3D is mainly on class-aware local grouping step, which iterates over all the classes to generate high-quality 3D proposals. However, in our approach, we use semantic threshold to select a point subset for each category, which can significantly reduce the computation usage. To achieve faster running speed, we also present a light-weight version with the larger voxel size (0.04m). With this modification, our model can be faster and still maintain a high performance. In addition, we further add the inference time comparison between our RoI-Conv module and other alternatives in Table 6. It can be seen that RoI-Conv pooling module is significantly more memory-and-time efficient than previous pooling operation. Hope it can be useful for the following two-stage methods.

Table 5: Performance comparison of latency and runtime memory on ScanNet V2 dataset. All methods are tested on same workstation.

| Method | 1-stage | 2-stage | total latency | memory | mAP@0.25 | mAP@0.5 |
|---|---|---|---|---|---|---|
| VoteNet [8] | 101.0ms | - | 101.0ms | 2,507MB | 58.6 | 33.5 |
| Group-free [6] | 153.1ms | - | 153.1ms | 3,678MB | 69.1 | 52.8 |
| FCAF3D [4] | 114.9ms | - | 114.9ms | 3,755MB | 71.5 | 57.3 |
| Ours(light) | 111.6ms | 12.3ms | 123.9ms | 2,947MB | 74.0 | 60.1 |
| Ours | 144.8ms | 34.5ms | 179.3ms | 3,544MB | **75.1** | **61.3** |

Table 6: Comparison with other RoI pooling approaches.

| RoI Method | mAP@0.25 | mAP@0.5 | memory | speed |
|---|---|---|---|---|
| PointRCNN | 73.65 | 57.83 | 8,054MB | 62.9ms |
| Part-A$^2$ | 74.01 | 58.89 | 6,540MB | 47.9ms |
| Ours-SA | 73.89 | 58.14 | 11,508MB | 45.5ms |
| Ours-SpConv | **74.50** | **60.31** | **2,468MB** | **34.5ms** |

| Table 7: Ablation study of feature shifting. | | |
| :---: | :---: | :---: |
| Feature Shifting | mAP@0.25 | mAP@0.5 |
| | 74.18 | 60.17 |
| ✓ | **74.50** | **60.31** |

| Table 8: Ablation study of loss weight. | | |
| :---: | :---: | :---: |
| $\beta_{rebox}$ | mAP@0.25 | mAP@0.5 |
| 1.0 | 74.29 | 60.15 |
| 0.5 | **74.50** | **60.31** |

## 2.3 More Ablation Studies

**The effect of feature offsets.** We provide the ablative studies of the feature shifting operation in our class-aware grouping module on ScanNet V2. We can observe in Table 7 that feature shifting is slightly better than the variant of non-shift. As discussed in VoteNet, to generate more reliable object representations, a MLP is used to transform seeds' features extracted from backbone to vote space, so that the grouped features can align with the voted points automatically.

**The effect of different loss weights.** As mentioned in §1.4, we simply set all the loss weights to 1.0 except for the bbox refinement $\beta_{rebox}$, which is adjusted to 0.5 for balancing the value of Stage-I box loss $\mathcal{L}_{box}$ and Stage-II refinement loss $\mathcal{L}_{rebox}$. Our method is not sensitive to loss weight and causes only minimal fluctuations (*e.g.* less than 0.3) as shown in Table 8.

**More possible combinations of the imporatant modules.** In Table 9, we list more results of combining different modules mentioned in the main paper including Semantic Prediction, Diverse Local Group, RoI-Conv and BiResNet. It can be seen that our well-designed modules can still boost the performance with various combinations, which shows the robustness and effectiveness of our method.

Table 9: Effect of Semantic Prediction, Diverse Local Group, RoI-Conv and BiResNet.

| Semantic Prediction | Diverse Local Group | BiResNet | RoI-Conv | mAP@0.25 | mAP@0.5 |
| :---: | :---: | :---: | :---: | :---: | :---: |
| | | | ✓ | 69.10 | 57.62 |
| ✓ | ✓ | | ✓ | 73.14 | 59.85 |
| | | ✓ | ✓ | 70.99 | 58.42 |
| ✓ | ✓ | ✓ | ✓ | **74.50** | **60.31** |

## 2.4 Quantitative Results

We provide the visualization of our prediction bounding boxes on ScanNet V2 and SUN RGB-D datasets. Please see Figure 1 for more qualitative results. Notably, our method can even accurately detect some miss-annotated objects in SUN RGB-D as in the bottom left of the figure.