# OpenReview forum: "CAGroup3D: Class-Aware Grouping for 3D Object Detection on Point Clouds"
_NeurIPS.cc/2022/Conference — NeurIPS 2022 Accept_

### Official Review · Reviewer_7kNx · 2022-07-04

**Rating:** 7
**Confidence:** 4
**Soundness:** 3 good
**Presentation:** 3 good
**Contribution:** 3 good

**Summary:**

This paper presents an approach for 3D detection from voxels. The authors propose to 2 stage detection pipeline that (1) first generates high quality box proposals by considering the spatial features of the scene. Compared to prior work, this stage also uses semantic information associated with each voxel. (2) Refines the object proposals and aggregates per-object features by a ROI pooling module. While prior work uses Set Abstraction (proposed in PointNet++) or max-pooling, the authors propose a new Sparse Abstraction operator. The final model is evaluated on indoor 3D detection on the ScanNet and SUN RGB-D datasets.

**Questions:**

+ How are the weights in Eqn (8) tuned? How sensitive is the model to these weights?
+ Did you try any simpler alternatives to the re-voxelization step? For example, rather than creating separate subsets, did you try feeding in the semantic probabilities of the classes into the aggregation module (which could be class-specific)?
+ Do you mean gIOU loss when you mention IOU loss in L237? Also, is the VoteNet baseline in Table 2 also trained with the IOU loss?
+ Are there particular classes for which this model is better than prior work? Thin/small objects? Since the authors spend effort in accounting for each object class’s size while aggregating features, I expect this design decision may work particularly well for some classes.

**Limitations:**

Yes.

**Strengths And Weaknesses:**

Strengths
+ The authors report the results averaged over 25 trials in each table. They account for both the training time stochastic behavior as well as inference time stochastic behavior in 3D detection models. This is important for follow-up work as it (a) empirically validates the current paper’s claims; (b) helps understand the robustness of each proposed component.
+ Strong empirical performance on indoor detection benchmarks. In particular, as Table 1 shows, the proposed model achieves a much higher mAP@0.5 than prior work.
+ The Class-aware proposal prediction is well motivated. In particular (L143), predicting a dense semantic map before creating the proposals is a good idea since it additionally supervises the feature backbone. I also liked that the authors directly use the 3D bounding boxes for this part rather than segmentation masks (L158).
+ The 3D backbone inspired by HRNet is a good choice for trading of resolution and compute. It also leads to good gains in performance (Table 2)

Weaknesses
+ The title suggests that the authors use point clouds directly. However, their implementation first converts the point clouds into voxels (L102). The title is misleading. Similarly, the introduction talks about challenges of using unordered point clouds which do not exist once point clouds are converted to voxels, and thus are not challenges either faced by this work or addressed by it. Please update the title/introduction.
+ The re-voxelization step introduced in Eqn (4) seems pretty involved - (a) it requires a threshold per class which is changed with a custom schedule during training. This seems like a non-trivial hyper parameter search. (b) It additionally requires re-voxelization using using a separate parameter \alpha. (c) Finally, it requires aggregation using a kernel k_alpha. I worry that this step will need re-tuning on different datasets.

---

> ### Author Response · Authors · 2022-08-02
> **Official Responses of Reviewer 7kNx**
>
> We sincerely thank the reviewer for providing thoughtful review and positive feedback. Below are our responses to the questions and suggestions raised by the reviewer.
>
> **R4-Q1: Update the title/introduction.**
> **R4-A1:** Thanks. We agree that the title and introduction are somewhat misleading. Our model takes the raw point clouds as input and leverages a learnable voxel feature encoder to voxelize the point clouds into 3D volumes, which can be processed by more efficient sparse convolution.
> We will follow the reviewer's suggestion and clarify this in the revised version.
>
> **R4-Q2: The non-trivial hyper parameter will need re-tuning on different datasets.**
> **R4-A2:** As for the concern about the hyper-parameter re-tuning, all the hyper parameters such as semantic threshold $\tau$ and scale factor $\alpha$ are shared across the different datasets without being re-tuned, which shows the great generalizability and robustness of our method. Although category average spatial dimension is dataset-specific, it is also easy to be obtained by making dataset statistics without being hand-designed.
>
> **R4-Q3: The weights in Eqn (8).**
> **R4-A3:** As mentioned in appendix, we simply set all the loss weights to 1.0 except for the bbox refinement $\beta_{rebox}$, which is adjusted to 0.5 for balancing the value of Stage-I box loss $\mathcal{L}_\text{box}$ and Stage-II refinement loss $\mathcal{L}_\text{rebox}$. Our method is not sensitive to these two reasonable loss weights (*e.g.*, 1.0 and 0.5) and causes only minimal fluctuations (*e.g.* less than 0.2) as shown below. We will ablate more loss weights in the revised version.
>
> | $\beta_{rebox}$ | mAP@0.25  | mAP@0.50  |
> |:---------------:|:---------:|:---------:|
> | 1.0             | 74.29     | 60.15     |
> | 0.5             | **74.50** | **60.31** |
>
> **R4-Q4: Any simpler alternatives to the re-voxelization step. Feeding semantic probabilities into aggregation module?**
> **R4-A4:** Thanks for your suggestions. We augment the backbone features with semantic probabilities and conduct ablation experiments on ScanNet V2.
>
> | Strategy               | mAP@0.25  | mAP@0.50  |
> |:----------------------:|:---------:|:---------:|
> | w/o                    | 70.99     | 58.42     |
> | semantic probabilities | 71.55     | 58.93     |
> | Ours                   | **74.50** | **60.31** |
>
> As shown in the above table, compared with the baseline that equips with all the components except for semantic predictions and diverse local grouping, simply adding voxel-wise semantic probabilities can also boost the performance. That indicates the semantic cues are really important for grouping based methods. However, our class-aware local grouping still outperforms this variant with a large gain. It can be explained that our model can not only explicitly leverage the strong semantic information to generate class-specific subsets for better guiding the network training, but also introduce the diverse locality and semantic consistency for solving the mis-grouping problem.
>
> **R4-Q5: IOU loss is gIOU loss? VoteNet baseline in Table 2 also trained with the IOU loss?**
> **R4-A5:** We use the naive IoU loss instead of gIoU loss by simply maximizing the IoU between the proposals and corresponding ground truths. To make a fair comparison, all the experiments in Table 2 are trained with the same loss objective, including the naive IoU loss adopted in $\mathcal{L}_\text{box}$ and $\mathcal{L}_\text{rebox}$.
>
> **R4-Q6: Are there particular classes for which this model is better than prior work?**
> **R4-A6:** Yes. Per-class results in appendix show that our model significantly performs better than prior works on tiny classes (*e.g.*, picture: +10.80 and +13.48 better than the SOTA in terms of mAP@0.25 and mAP@0.5 on ScanNet V2), which demonstrates the effectiveness of our local grouping strategy.

---

### Official Review · Reviewer_QGEb · 2022-07-10

**Rating:** 5
**Confidence:** 4
**Soundness:** 2 fair
**Presentation:** 3 good
**Contribution:** 2 fair

**Summary:**

The authors introduced a novel two-stage 3D point cloud framework in indoor environment. In the previous bottom-up framework, the authors firstly modified the class-agnostic local grouping scheme, which may fail in clustering the object close to different categorical object. Instead of grouping class-agnostic manner as VoteNet, the proposed method grouped the voxels with the semantic consistency from the additional semantic prediction branch. Furthermore, the authors proposed an efficient ROI pooling module to compensate the miss-classified semantic prediction in the second stage. In the experimental section, the authors provided experiments on ScanNet V2 and SUN RGB-D benchmarks to demonstrate the proposed methods succeeds in grouping the objects within the same group and shows the better performance compared to existing class-agnostic methods.

**Questions:**

Feedbacks of weeknesses for validity of RoI pooling and opinion to reinforce other contributions.

**Limitations:**

I mentioned all comments including reasons and suggestions in the above sections. I recommend that the author will provide all the concerns, and improve the completeness of the paper (there is few grammatical error).

**Strengths And Weaknesses:**

[+] First of all, the motivation of the paper seems to be meaningful and pragmatic for indoor 3D object detection, because different types of objects may be located in a short distance in the indoor environment unlike the outdoor environment.

Despite of lacks of technical novelties, the authors well modified the existing class-agnostic model to solve the desired problem. The backbone model is selected to consider the geometric characteristic of the indoor object target, and the pooling module is designed for sparse convolution to handle limitations as computational overhead and geometric information loss.

As the detection paper, the overall framework is well-described, and specifically the details for reproducing and understanding the contribution such as target assignment, handling unique set, comparisons on different kernel size etc. are well presented.


[-] One concern is the proposed RoI pooling module. In sparse convolution, there is a hand-crafted parameter such as voxel size, and according to this factor, geometric loss occurs like existing pooling methods.

Another concern is the role of the diverse local grouping. In Table 2, this factor has the greatest impact on performance improvement compared to other factors. Among all the contributions, this factor does not look much different from operating a class-agnostic model for each class efficiently. Therefore, I recommend that the authors should provide the opinion to reinforce other contributions.

---

> ### Author Response · Authors · 2022-08-02
> **Official Responses of Reviewer QGEb**
>
> We sincerely thank the reviewer for providing thoughtful review and positive feedback. Below are our responses to the questions and suggestions raised by the reviewer.
>
> **R3-Q1: Validity of RoI pooling.**
> **R3-A1:** Sorry for this confusion, we will add more clarification in the revised version. As for the loss of geometric details, our RoI-Conv module doesn't equip with re-voxelization or pooling steps since the RoI-Conv module directly operates voxel features from the backbone. Note that the voxel size is small enough (0.04m) so that it can well capture the fine-grained surface geometry and achieve impressive performance on tiny objects (*e.g*., picture). Actually, our model only performs voxelization in two steps: 1) At the beginning of feeding the input into 3D voxel backbone, the point clouds are first converted to regular 3D voxels with the voxel size of 0.02m, which is fine enough for detecting coarse-grained 3D bounding boxes. 2) In the diverse local grouping step, our model re-voxelizes the semantic subsets respectively by average pooling operation to generate class-specific 3D voxels. Notably, it is class-aware so that enough geometric information can be adaptively retained for detection. The entire operation is also differentiable and the obtained voxel features can be easily adapted to original point features.
>
> **R3-Q2: The role of the diverse local grouping.**
> **R3-A2:** Locality is one of the two crucial inductive biases we introduced to grouping algorithm, which is class-dependent and diverse among the different classes. However, previous grouping methods of 3D object detection are usually in a class-agnostic manner, which leads to the mis-grouping problem, *e.g.,* partial or over coverage of the object surfaces. Our diverse local grouping operation tries to address this problem. We first re-voxelize the semantic subsets with class-specific voxel size (*w.r.t*, the average spatial dimension of each category) and then group them by sparse convolutions with the same kernel size individually. Thus the smaller classes are preferred to be aggregated with smaller local regions and more detailed geometric representations, and the larger classes vice versa. It's novel and reasonable for the bottom-up 3D detection framework.
> To ablate its effectiveness, we compare our model with the variant that directly operates the class-agnostic model on each class subset without re-voxelization. As shown in 2$^{nd}$ and 3$^{rd}$ row of Table 2, our diverse local grouping achieves much better performance, *i.e.*, 69.24 $\rightarrow$ 72.10, 54.05 $\rightarrow$ 57.07 on mAP@0.25 and mAP@0.5, which demonstrates its superiority. We sincerely thank the reviewer again and will reinforce other contributions such as other crucial inductive biases (semantic consistency), strong 3D backbone, and the efficient fully convolutional aggregation / 3D pooling module in the revised version. Please feel free to let us know if we misunderstood this question.
>
> **R3-Q3: Improve the completeness of the paper (there is few grammatical error).**
> **R3-A3:** Thanks. We will carefully polish our writing and fix the grammatical errors in the revised version.

---

### Official Review · Reviewer_uwGH · 2022-07-10

**Rating:** 5
**Confidence:** 3
**Soundness:** 3 good
**Presentation:** 3 good
**Contribution:** 3 good

**Summary:**

Existing 3D indoor object detection usually abandons semantic consistency within the same group and ignores diverse locality among different categories. To tackle this issue, the authors propose a novel class-aware proposal generation strategy that considers class-specific local groups. Besides, the authors propose an RoI pooling module to revisit the missed surface voxel features due to semantic segmentation errors. Their approach outperforms state-of-art methods on two challenging indoor datasets.

**Questions:**

- I am not sure about the effect of feature offsets. What is the difference between features extracted from the backbone and features added by feature offsets? Please demonstrate the design goal of feature offsets.
- As the set of neighboring input voxels is dependent on RoI-specific points, sparse abstraction is a time-expensive process. Please compare the inference time between the proposed method and existing methods in the parts of the experiment.
- In Table 2, it is important for a reader to see all possible combinations of important modules (e.g. class-aware grouping and RoI-Conv) and see their potential influences.  I wonder if a better backbone can alleviate the improvements of local grouping.
- There is no qualitative analysis in the main paper. It is hard for a reader to judge what are the potential improvements.

**Limitations:**

Yes.


**Strengths And Weaknesses:**

Strengths:
+ The authors utilize a voxel feature encoding with a class-specific 3D voxel size to obtain the predicted vote voxels and generate class-specific geometric features based on them.
+ The authors propose RoI-Conv pooling, which effectively recovers the missed surface voxel features due to the semantic segmentation errors
+ The proposed method achieves state-of-art results with a remarkable gain on ScanNet V2 and SUN RGB-D.

Weaknesses:
- The idea of using semantic predictions and geometric shifts to get class-aware clusters is not novel, which has already been used in previous work such as SoftGroup.

---

> ### Author Response · Authors · 2022-08-02
> **Official Responses of Reviewer uwGH**
>
> We sincerely thank the reviewer for providing thoughtful review and positive feedback. Below are our responses to the questions and suggestions raised by the reviewer.
>
> **R2-Q1: The idea of using semantic predictions and geometric shifts to get class-aware clusters is not novel, such as SoftGroup.**
> **R2-A1:** SoftGroup is a valuable work in the 3D instance segmentation community and will be involved in our revised version. However, our approach is very different from the previous voting-based semantic clustering methods in the following aspects: 1) Existing semantic clustering algorithms are usually adopted in 3D instance segmentation, which aims to explicitly assign each point to its corresponding instance accurately. But our model is designed for 3D object detection and focuses on efficiently introducing two crucial inductive biases, *i.e.*, semantic consistency and diverse locality, to grouping step for implicitly aggregating high-quality object representations and generating reliable coarse-grained bounding boxes. The strong results in Table 2 verify the effectiveness of the proposed strategy. 2) Our grouping module is a fully sparse convolutional approach, which automatically aggregates the object surface points with pure sparse convolutions and avoids the hand-crafted clustering operations. Thanks to the well-optimized sparse convolution, our aggregation model achieves 3$\times$ faster than the fastest semantic clustering algorithm (*i.e.*, softgroup).
>
> **R2-Q2: The effect of feature offsets.**
> **R2-A2:** To make a fair comparison, we follow the widely used feature shifting operation proposed in VoteNet [25]. Additional ablative studies of this step are provided below.
>
> | Feature Shifting | mAP@0.25    | mAP@0.50   |
> |:----------------:|:-----------:|:----------:|
> |                  | 74.18       | 60.17      |
> |  $\checkmark$    | **74.50**   | **60.31**  |
>
> We can observe that feature shifting is slightly better than the variant of non-shift. As discussed in VoteNet, to generate more reliable object representations, a MLP is used to transform seeds’ features extracted from the backbone to vote space, so that the grouped features can align with the voted points automatically.
>
> **R2-Q3: Compare the inference time between the proposed method and existing methods.**
> **R2-A3:** Thanks for the valuable suggestion. Table 5 will be updated with inference time.
>
> | RoI Method       | mAP@0.25  | mAP@0.50  | memory      | inference time |
> |:----------------:|:---------:|:---------:|:-----------:|:--------------:|
> | PointRCNN [28]   | 73.65     | 57.83     | 8,054MB     | 62.9ms         |
> | Part-A$^{2}$ [30]| 74.01     | 58.89     | 6,540MB     | 47.9ms         |
> | Ours-SA          | 73.89     | 58.14     | 11,508MB    | 45.5ms         |
> | Ours-SpConv      | **74.50** | **60.31** | **2,468MB** | **34.5ms**     |
>
> We can find that the RoI-Conv pooling module is significantly more memory- and time- efficient than previous pooling operations. Hope it can facilitate the two-stage 3D object detection community.
>
> **R2-Q4: All possible combinations of important modules. If a better backbone can alleviate the improvements of local grouping.**
> **R2-A4:** Totally agree. Due to limited space, we only focus on the most significant combinations. The results of all possible combinations will be added in the revised version.
>
> |Semantic Prediction|Diverse Local Group|BiResNet    |RoI-Conv    | mAP@0.25 | mAP@0.50|
> |:-----------------:|:-----------------:|:----------:|:----------:|:--------:|:-------:|
> |                   |                   |            |$\checkmark$|69.10     |57.62    |
> |$\checkmark$       |$\checkmark$       |            |$\checkmark$|73.14     |59.85    |
> |                   |                   |$\checkmark$|$\checkmark$|70.99     |58.42    |
> |$\checkmark$       |$\checkmark$       |$\checkmark$|$\checkmark$|**74.50** |**60.31**|
>
> As for the concern of better backbone, we list parts of the results and find that local grouping can still boost the performance, even with a better backbone (BiResNet).
>
> **R2-Q5: No qualitative analysis.**
> **R2-A5:** Thanks. We will move some qualitative analysis from appendix to main paper and add more additional analysis. For example, per-class results in appendix show that our model is significantly better than prior works on tiny classes (*e.g.*, picture: +10.80 and +13.48 better than the SOTA in terms of mAP@0.25 and mAP@0.5 on ScanNet V2), which demonstrates the effectiveness of our local grouping strategy.

---

### Official Review · Reviewer_nq6L · 2022-07-12

**Rating:** 6
**Confidence:** 4
**Soundness:** 3 good
**Presentation:** 3 good
**Contribution:** 2 fair

**Summary:**

The paper addresses the problem of 3D object detection in point cloud for indoor scenes (scannet and sun rgbd). The idea is based on voting-based methods: each point predicts its center -> clustering -> classification. The method proposes to classify points first, before center prediction and clustering. They claim contributions in the following:

- Novel "class-aware" 3D proposal strategy as discussed above.

- A refinement stage that re-voxelizes according to each point's classes.

- Good results on scannet and sun rgb-d.

The results do seem strong. And both claimed novelties contributed to the improvements. I'll recommend a weak accept for now. Might change opinions based on peers' reviews.


**Questions:**

N/A

**Limitations:**

There is one sentence. What do the authors mean by diversity localities?

**Strengths And Weaknesses:**

Strengths:

- Strong results on both datasets as mentioned. Also good ablation on table 2 to demonstrate the improvements from the claimed novelties.

- Good writing.

Weaknesses:

- Only somewhat sufficient novelty: I don't feel that reordering the steps of a voting-based method constitutes to sufficient novelty for this venue, but maybe only barely together with the 2nd-stage re-voxelization idea. A 2nd-stage detector is followed by initial points segmentation is not really new either. See [RSN: Range Sparse Net for Efficient, Accurate LiDAR 3D Object Detection]

---

> ### Author Response · Authors · 2022-08-02
> **Official Responses of Reviewer nq6L**
>
> We sincerely thank the reviewer for providing thoughtful review and positive feedback. Below are our responses to the questions and suggestions raised by the reviewer.
>
> **R1-Q1: Don't feel reordering the steps of voting-based method constitutes to sufficient novelty, but maybe only barely together with the 2nd-stage re-voxelization idea.**
> **R1-A1:** We modify VoteNet [25] and adapt it to be compatible with voxel representations for the succeeding class-aware local grouping module. Our main motivation is to solve the mis-grouping problem caused by class-agnostic grouping in VoteNet, which is crucial for bottom-up 3D object detectors in cluttered indoor scenes where the objects of different classes are distributed closely. Our class-aware local grouping tries to address this limitation by introducing two reasonable inductive biases, *i.e.* semantic consistency and diverse locality among different categories. The strong results demonstrate the novelty and effectiveness of our method. Combining the class-specific re-voxelization step with voting approach is a simple yet effective way to achieve the above ideas.
>
> **R1-Q2: A 2nd-stage detector is followed by initial points segmentation is not really new either.**
> **R1-A2:** RSN only performs foreground point segmentation for detecting 3D objects more efficiently. Our approach and motivation are very different from it. We perform voxel-wise semantic predictions on all categories rather than only foreground, which aims to assign the points to their respective semantic subsets and achieve class-aware local grouping more efficiently.
>
> **R1-Q3: What do the authors mean by diverse localities in limitation discussion?**
> **R1-A3:** Sorry for this confusion, we will clarify it and add more descriptions in the revised version. Locality is one of the most important inductive biases of the grouping-based 3D object detection framework. This paper mainly focuses on the inter-category locality, which is class-specific and diverse among the different classes,  but ignores the intra-category discriminations. Due to the incompleteness of the point cloud and the scale variance within the classes, the object spatial dimension of the same class is also variational, which leads to the diverse intra-category locality. Although our grouping algorithm can implicitly handle this problem to some degree by the learnable convolutional aggregation module, it is still an open problem and will be studied in the future.

---

### Meta-Review · Area_Chair_cNrL · 2022-08-25

**Recommendation:** Accept
**Confidence:** Certain

**Metareview:**

4 expert reviewers suggest acceptance, based mostly on a strong evaluation section that shows good improvements over previous methods. Novelty of the method is deemed sufficient and well ablated. Overall seems like a good quality paper, although a tiny bit on the incremental side, but enough for recommending acceptance.

**Award:**

No

---

### Decision · Program_Chairs · 2022-09-14

Accept